



# Tsunami scenario triggered by submarine landslide offshore of northern Sumatra Island and its hazard assessment

Haekal A. Haridhi[1,9,10], Bor Shouh Huang[2, *], Kuo Liang Wen[3], Arif Mirza[4], Syamsul Rizal[1,9], Syahrul Purnawan[1,9], Ilham Fajri[12], Frauke Klingelhoefer[5], Char Shine Liu[4], Chao Shing Lee[6], Crispen R. Wilson[7], Tso-Ren Wu[8], Ichsan Setiawan[1,9,11] and Van Bang Phung[2]

[1]Department of Marine Sciences, Faculty of Marine and Fisheries, Universitas Syiah Kuala, Banda Aceh, Indonesia.
[2]Institute of Earth Sciences, Academia Sinica, Taipei, Taiwan.
[3]Department of Earth Sciences, National Central University, Taoyuan, Taiwan.
[4]Ocean Center, National Taiwan University, Taipei, Taiwan.
[5]Ifremer, Department of Marine Geosciences, Plouzané, France.
[6]Institute of Geosciences, National Taiwan Ocean University, Keelung, Taiwan.
[7]ENC, Washington, D.C., USA.
[8]Graduate Institute of Hydrological and Oceanic Sciences, National Central University, Taoyuan, Taiwan.
[9]Research Center for Marine Sciences and Fisheries, Universitas Syiah Kuala, Banda Aceh, Indonesia.
[10]Tsunami and Disaster Mitigation Research Center, Universitas Syiah Kuala, Banda Aceh, Indonesia.
[11]Graduate School of Mathematics and Applied Sciences, Universitas Syiah Kuala, Banda Aceh, Indonesia.
[12]Department of Capture Fisheries, Marine and Fisheries Polytechnic Aceh, Great Aceh, Indonesia.

*Correspondence to*: Bor Shouh Huang (hwbs@earth.sinica.edu.tw)

**Abstract.** Near the northern border of Sumatra, the right-lateral strike-slip Sumatran Fault Zone splits into two branches and extends into the offshore, as revealed by seismic sounding surveys. However, due to its strike-slip faulting characteristics, the Sumatran Fault Zone's activity is rarely believed to cause tsunami hazards in this region. According to two reprocessed reflection seismic profiles, the extended Sumatran Fault Zone is strongly associated with chaotic facies, indicating that large submarine landslides have been triggered. Coastal steep slopes and new subsurface characteristics of submarine landslide deposits were mapped using recently acquired high-resolution shallow bathymetry data. Slope stability analysis revealed some targets with steep morphology to be close to failure. In an extreme case, an earthquake of $M_w$ 7 or more occurred, and the strong ground shaking triggered a submarine landslide off the northern shore of Sumatra. Based on a simulation of tsunami wave propagation in shallow water, the results of this study indicate a potential tsunami hazard from a submarine landslide triggered by the strike-slip fault system. The landslide tsunami hazard assessment and early warning systems in this study area can be improved on the basis of this proposed scenario.

## 1 Introduction

The most distinct tectonic feature of Sumatra (Fig. 1a) is a strike-slip fault called the Sumatran Fault Zone (SFZ) with a length of approximately 1900 km that stretches from Sunda Strait to Banda Aceh (Sieh and Natawidjaja, 2000). Numerous





large earthquakes have occurred along this long fault zone; the largest with a magnitude of 7.7 (Sieh and Natawidjaja, 2000).

Near the northern tip of Sumatra (Aceh Province), the SFZ divides into two branches: the Aceh and Seulimeum faults (Fig. 1b). These faults cross the northern end of Sumatra Island and extend into the Andaman Sea (Fernández-Blanco et al., 2016; Ghosal et al., 2012; Sieh and Natawidjaja, 2000). The most recent earthquake at the Seulimeum fault was recorded on 2 April 1964, with $M_w$ 7, and it caused severe damage at Krueng Raya (Sieh and Natawidjaja, 2000). After the 2004 Sumatra–Andaman earthquake and tsunami, the source rupture region was studied intensively; in particular, research has focused on

the western side of the island and areas near the epicenter. By contrast, the inland SFZ has rarely been studied. The absence of high-resolution geophysical data at these segments has resulted in limited knowledge about the seismic activities of these two extensions and the corresponding risk of hazards.

Analysis of the $M_w$ 7.0 Haiti earthquake on 12 January 2010 revealed that an earthquake with strike-slip faulting can produce a significant tsunami. Typically, a strike-slip fault movement is not associated with uplift of the sea floor or tsunami

generation. However, a combination of other factors can trigger a tsunami. For the Haiti earthquake, the tsunami waves seem to have been caused by coastal failure landslides (Poupardin et al., 2020 and references therein). Satellite images and ground photos reveal changes in the coastline following the earthquake (Hornbach et al., 2010). The Haiti earthquake is not unique. On 28 September 2018, a large tsunami hit the city of Palu following the $M_w$ 7.5 Sulawesi earthquake in Indonesia. This event also occurred along a strike-slip fault. A tsunami of that size is unlikely to have been generated through earthquake

rupturing alone. The tsunami is thought to have been caused by underwater and subaerial landslides triggered by the earthquake (Gusman et al., 2019). The complex bathymetry of the Palu Bay may have also contributed to the generation of the tsunami (Socquet et al., 2019). Other well-known tsunamis, such as the 1998 Papua New Guinea abnormal tsunami (Heinrich et al., 2001; Kawata et al., 1999; Tappin et al., 1999) and the 22 December 2018 tsunami at Sunda Strait caused by a flank collapse of the Anak Krakatau Volcano (Heidarzadeh et al., 2020; Muhari et al., 2019; Patton et al., 2018; Syamsidik

et al., 2020), were also induced by earthquake-triggered submarine landslides (Ye et al., 2020).

An earthquake with a strike-slip fault rupture could also trigger a landslide and induce a tsunami offshore of northern Sumatra. In this study, to investigate the potential tsunami hazard at the northern tip of Sumatra, seismic reflection data were used to identify evidence of past submarine landslides. We collected detailed shallow bathymetric data of the area beyond the coast. This high-resolution bathymetric data were used to identify the fault traces and to evaluate the possibility of slope

failure along the continental slope. The possibility of a submarine landslide triggered by earthquake shaking was examined through an analysis of the continental slope stability, and a tsunami caused by the combination of the earthquake and the resulting submarine landslide was simulated. The results indicated the characteristics of a potential landslide-induced tsunami and its potential damage. A possible tsunami early warning plan for hazard reduction is also discussed in this paper.



## 2 Tectonic setting of the northern SFZ

The oblique subduction of the Australian plate below the Sunda plate is compensated by right-lateral strike-slip faulting at the Sunda plate along Sumatra (McCaffrey, 1992). Tectonic processes in the present-day Sumatra region are controlled by three major fault systems: the megathrust fault along the Sunda Trench (reverse fault), the Mentawai fault (right-lateral strike-slip) (Barber and Milsom et al., 2005; Berglar et al., 2017; Moore et al., 1980), and the SFZ (right-lateral strike-slip) (Fig. 1a). The fault trace of the SFZ is associated with a series of valleys along the mountain chain (Sieh and Natawidjaja,

2000). The linkage of the SFZ from south to north is not smooth; the fault divides into several segments (Newcomb and McCann, 1987; Sieh and Natawidjaja, 2000). The SFZ segmentation also results in the segmentation of earthquakes and serves as a rupture barrier, decreasing earthquake magnitudes (Barber and Milsom et al., 2005; Sieh and Natawidjaja, 2000). At the northern corner of Sumatra, the SFZ divides into two branches named the Aceh and Seulimeum faults. Both faults have been reported to extend northerly into the Andaman Sea floor (Fernández-Blanco et al., 2016; Ghosal et al., 2012; Sieh

and Natawidjaja, 2000). A recent detailed investigation of the Aceh and Seulimeum fault geometries revealed a complex fault system for both faults (Fernández-Blanco et al., 2016). The Aceh fault has fold train features that evolve as splay contractional structures of the overall strike-slip system. The Seulimeum fault divides into two branches, and a long valley is formed at the northern end (Fig. 1b). The two Seulimeum fault traces were identified by (Fernández-Blanco et al., 2016) as "Set A" and "Set B"; here, we refer to them as Seulimeum fault 1 (S1) and Seulimeum fault 2 (S2), respectively; the traces

are indicated in Fig. 1b. In the region between the Aceh and Seulimeum faults, no fault trace indicating interactions between these two main branches of the SFZ was observed. However, because no geophysical data are available in this area, the existence of a fault trace buried by deep sediment cannot be ruled out. The slip rates along the Aceh and Seulimeum faults have not been reported. However, an offset of approximately 20 to 21 km in the nearby segment of the SFZ in the Aceh region has been observed within the past several million years (Sieh and Natawidjaja, 2000), and the Banda Aceh

embayment extrudes to the northwest at a rate of $5 \pm 2$ mm y$^{-1}$ (Genrich et al., 2000). Both observations indicate the activity of the Aceh and Seulimeum faults.

## 3 Collected data and analysis methods

### 3.1 Single-channel seismic reflection data

From 1991 to 1992, single-channel seismic (SCS) reflection data have been collected along the western margin of Sumatra

(Malod and Kemal, 1996). These collected data include five seismic profiles offshore north of Aceh. In this study, we selected two profiles that cross the Seulimeum faults (i.e., SUMII-32 and SUMII-33; Fig. 1b) for further analysis. These data were originally recorded on paper prints. Those paper recordings were scanned and converted to digital images. All seismic traces were digitized and converted into the SEG-Y format for reprocessing. In the absence of any velocity information, these data were migrated using a water velocity of 1500 m s$^{-1}$ to remove the effects of seafloor scattering. The reprocessed



seismic profiles are presented in Figs. 2 and 3. The seismic profile SUMII-32 crosses over the northward extension of fault S2 (Fig. 1b). This profile is a short (26.5 km) seismic profile close to the coast. On this profile, the location of fault S2 is clearly visible in the seismic section. This fault trace depicts a near-vertical fault plane with a positive flower structure, indicating fault activity (Fig. 2b). A subsidence sequence marked by fan-shaped sediments is visible in the northeast section of the profile (Fig. 2c) and may indicate an extension regime of the back-arc basin. Figure 3 presents the seismic reflection

SUMII-33, which is located parallel to the northern coast of Banda Aceh and perpendicular to faults S1 and S2 (Fig. 1b); the figure depicts a system of shear faults that dip to the southwest and normal to the northeast on the western part of the profile (Fig. 3b). All of the faults are close to the surface and have a surface obstruction. These traces indicate recent activity in this shear fault system (Fig. 3).

### 3.2 Community-based bathymetric survey data

The Community-Based Bathymetric Survey (CBBS) data used in this study comprise fishing boat track records from GPS sounders with data logging devices. The data include the date, time, depth, sea surface temperature (SST), boat speed, heading, and geographical position (Haridhi et al., 2016; Rizal et al., 2013). Data were collected from 45 local fishing boats in the northern Sumatra area, which had installed sounder equipment due to participation in a project supported by the Asian Development Bank (ADB) as an effort to rehabilitate the traditional fishing community after the mega earthquake and

tsunami disaster on December 26, 2004 (Wilson and Linkie, 2012). The CBBS data collection was from June 2007 to May 2009 (23 months). A total of 6,170,648 data points from 922 data sets were collected. The collected fishing boat tracks were employed to construct high-spatial-density bathymetry with $20 \times 20$-m$^2$ grid spacing. To unite the bathymetric features and land features, topographic data from the Badan Informasi Geospasial of Indonesia (i.e., both provided topographic data names DEMNAS and bathymetric data names BATNAS with 0.27-arcsecond and 6-arcsecond grid resolution, respectively)

were used (BIG, 2019). BATNAS was resampled to match the grid resolution of CBBS bathymetry, whereas the original data resolution of DEMNAS was used. The reconstructed topography is presented in Fig. 4. The white dashed line in Fig. 4 indicates the boundary location of CBBS and BATNAS datasets with different resolution properties. The map covers the northern corner of Sumatra Island and the two branches of the SFZ (Aceh and Seulimeum faults), as indicated in Fig. 1b. Figure 5 presents a three-dimensional (3D) topographic view of bathymetry at locations near the continental slope, as

indicated in Fig. 4.

The Banda Aceh coast near the shallow water area of Fig. 4 is extruded toward the northeast by a potential shear fault system between the Aceh and S1 faults. Tectonic movement along the main fault system may induce the movement of these shear faults, as indicated by the contour pattern of these shear faults with the southwest–northeast orientation (Fig. 4). The contour line has a step over at the west side of the shear fault, and the edge portion of these shear faults has an increased risk of

collapse during an earthquake. Four shear faults (f1, f2, f3, and f4 in Fig. 4) with left-lateral slips can be identified in conjunction to the right-lateral slip movement of the main fault system. A distinct difference in water depth is observed off the northern shore of Sumatra. Some identified scarps located at the slope close to f1 and f2 can be clearly observed, and the





location of a possible historical landslide is marked along f3 (Fig. 5). At least one mound-shaped submarine landscape is located on the plain at approximately 2226 m from the slope close to f2 (Fig. 5). The highest-resolution CBBS bathymetry is

limited for shallow water areas; however, these shear faults can be identified in the seismic reflection line SUMII-33 (Fig. 3), and they extend continuously northward from the shallow waters. An interpretation of both the SCS seismic profile in Fig. 3b and the shallow bathymetry in Fig. 4 reveal that these four shear faults (f1, f2, f3, and f4 in Fig. 4) accompanied by normal slips have a negative flower structure.

### 3.3 Slope stability analysis and input parameter assessment

An analysis was performed based on the assumption that earthquake-induced shaking may trigger submarine landslides on the unstable continental slope. Scoops3D, a computer program for evaluating the slope stability throughout a digital landscape represented by a digital elevation model, was selected for the analysis (Reid et al., 2015). The program calculates the slope stability through limit-equilibrium methods, which estimate the shear resistance on the trial surface of a potential failure mass in 2D (vertical slices) and 3D (vertical columns) (Duncan, 1996; Reid et al., 2015). Scoops3D computes a factor

of safety (FS) for a given trial surface by using the moment equilibrium. In all limit-equilibrium methods, the FS is defined as the ratio of the average shear resistance (strength) to the shear stress required to maintain a limiting equilibrium along a predefined trial surface. In Scoops3D, Bishop's simplified method can be used to determine the normal force acting on the slip surface by first computing the force equilibrium in the vertical direction on the base of each slice (Bishop, 1955). This method has been demonstrated to calculate FS accurately (Reid et al., 2015). Typically, FS > 1 is considered as representing

stability, and FS ≤ 1 represents instability.

Following the evaluation of the seismic profile and shallow bathymetry, the continental slope edges were found to be associated with the structure of the active fault; an example is presented in Fig. 5. Earthquake shaking is a critical factor for generating horizontal and vertical ground motion. This vibration causes both shear and normal stresses in the sediment. Horizontal acceleration has the highest contribution to the shear stress and can drive sediment failure near continental slope

edges (Hampton et al., 1996). To represent the effects of ground acceleration from an earthquake, Scoop3D models typically assume that earthquake loading is uniform (Reid et al., 2015); this assumption was also used in this study. However, in the absence of any supporting information other than detailed CBBS shallow bathymetric grid, other data such as the subsurface condition (i.e., cohesion and internal friction) were assumed in accordance with the findings of Dugan and Flemings (2002). Lee and Edwards (1986) examined the seismic active offshore margins of California and southern Alaska and suggested that

pseudostatic acceleration between 0.13 and 0.14 g (gravity) corresponds to the transition from stable to failed sediment; that is, such acceleration causes sediment failure. Thus, we selected 0.14 g as the earthquake loading in Scoop3D. The detailed parameters used in the Scoop3D slope stability analysis are summarized in Table 1. Locations with low identified FS values that are collocated with or close to a fault were candidates for slide locations in the tsunami generation model.


### 3.4 Simulation of tsunami wave propagation from earthquake and landslide sources

The Cornell multi-grid coupled tsunami model (COMCOT) (Liu et al., 1995; Wang, 2009) is a computer program applied for performing tsunami simulations. It simultaneously calculates the tsunami wave propagation and the inundation at coastal zones. In this study, the nonlinear shallow water equation was used to construct COMCOT. COMCOT can calculate the tsunami propagation from earthquake sources, submarine landslides, and both phenomena. As described by Wang and Liu (2006), COMCOT has been widely used to model tsunamis generated by earthquakes and the mega earthquake and tsunami

on December 26, 2004. To construct earthquake-source tsunami simulation input parameters, we used the magnitude scaling relationship of Wells and Coppersmith (1994) to convert the magnitude into the strike-slip fault area (RA) of an earthquake. Based on this relationship, we defined the RA, the subsurface rupture length (RL), and the rupture width (RW) from the moment ($M_0$) of a given earthquake. To calculate the average displacement ($\overline{D}$) across the fault surface, we used the seismic moment equation of Hanks and Kanamori (1979).

COMCOT can also be used to simulate tsunamis caused by landslides (Liu et al., 1995; Wang, 2009). In the landslide-generated tsunami simulation, an underwater slide of a rigid body along a particular downslope path is considered a tsunami source (Watts et al., 2003). Typically, modeling the time evolution of an actual landslide with seafloor changes requires substantial computations involving the detailed knowledge of local marine geological features and the landslide's triggering mechanism. To use COMCOT to model a landslide source, input parameters including the landslide mass position (c), length

(l), width (w), thickness (h), and the slope angle along its sliding path (φ) must be defined. In this study, Manning's relative roughness $n$ was set to 0.02 in accordance with the assumption that the continental shelf sediments were primarily composed of mud and silt (Lee and Edwards, 1986).

## 4 Analysis and results

### 4.1 Evidence of paleo-landslides

As presented in Fig. 1b, the seismic reflection SUMII-33 (Fig. 3) is a long profile located in sea between Sumatra Island and Weh Island, and it is nearly perpendicular to the extending fault traces of S1 and S2. In the western part of SUMII-33, chaotic facies at 17.5–20 km along the profile is clearly visible (Fig. 3b), and its location is at the northward extension of the four-shear fault zone, as identified by CBBS bathymetry (Figs. 4 and 5). The thickness of this chaotic facies is approximately 0.2 s two-way-time (TWT). If we assume that the seismic wave velocity of marine sediment is approximately 2000 m s$^{-1}$, the

thickness of this chaotic facies is approximately 400 m. In the eastern part of SUMII-33, large chaotic facies is observed (interpreted as mass transport deposit [MTD] facies) and is marked by the yellow dashed line in Fig. 3c. Similar thin-layer chaotic facies is also observed on the near coastal line of the short seismic profile SUMII-32 (Figs. 1b and 2). The chaotic facies is observed at the slope of the SUMII-32 profile and may be related to the downslope turbidity or gravity flow. A fan-shaped sediment is clearly observable on the abyssal plain near the continental slope edge and may indicate an extensional



regime of the back-arc basin (Fig. 2c). Some similar chaotic facies below these fan-shaped sediment sequences at a distance of 12,500 m and 2.4 s TWT are difficult to distinguish because the reflector amplitudes are too low, limiting our capability to interpret the profile (Fig. 2c). Furthermore, on this seismic profile, the S2 fault has a positive flower structure, indicating fault activity (Fig. 2b). These observations imply that a submarine landslide previously occurred in this area and could be triggered by a fault rupture. However, the precise landslide site along the S1 and S2 faults is difficult to identify due to the

low resolution of the obtained seafloor morphology data (Fig. 4). The low-resolution bathymetry data of the seismic survey area limited the identification of any evidence of scarp- or mound-type structures for the evaluation of possible submarine landslide sites.

## 4.2 Stability evaluation of seafloor morphology

The CBBS shallow bathymetric map (Figs. 4 and 5) has the highest-resolution seafloor morphology in comparison to any

other available bathymetric data set for shallow water in this region. The average angles along the continental shelf, continental slope, and abyssal plain calculated by Scoop3D are between 0° to 5°, 6° to 30°, and 0° to 15°, respectively (Fig. 6a). The abyssal plain typically has a low slope angle; however, some areas of the plain have substantially larger slope angles of 10°–30°, as calculated by Scoop3D (marked as green circles in Fig. 6a). These unusual high slope angles may correspond to the location of topographic high or low points; these morphological features are either submarine mountain or

deeper portions of the abyssal plain. However, these unusual high slope angles were detected at regions of the plain with sparse data; thus, the slope angle results may be unreliable. More reliable slope analysis results were obtained from regions within the CBBS survey area; only these results were used in the following analysis.

As presented in Figs. 4 and 5, two river mouths are the input source of sediment along this continental shelf; they are located on the Banda Aceh plain (see Figs. 4 and 5, text label: Aceh and Lamnyong rivers). These thick sediments may provide

support for extending the continental shelf. The shape of the continental shelf is further modified by the activity of the shear faults (e.g., f1 to f4 in this area) below the shelf. The spatial instability of the seafloor can be further evaluated using the FS index (Reid et al., 2015). Submarine landslides may occur in regions with low FS values. The slope stability analysis results of Scoop3D are presented in Fig. 6b. Areas with anomalously low FS values ≤2 are located at the continental slope offshore of Banda Aceh. The regions with low FS values and collocated with the four shear faults (f1 to f4) were identified as

locations for further tsunami simulations (marked as locations 4 to 7 in Fig. 6c). The sediment deposited above the area of the shear faults across the continental slope may increase the scale of landslides during earthquake shaking. Two other sites with FS values and collocated with the faults (S1 and S2) were also identified as possible submarine landslide locations, and they are marked as locations 1 and 3 in Fig. 6c. Another area with changes in the FS value near the coast is located offshore north of Krueng Raya; the FS drastically changes from 4 to 2 within a distance of 2 km. This location is also a submarine

landslide location for the tsunami simulation (location 2 in Fig. 6c). Figure 6b presents that the FS values along the continental slope (marked by a solid white circle) are stable and high. This location was identified as a previously failed



section of the continental slope along f3; this failure can be clearly observed in Fig. 5. This phenomenon further confirms our interpretation of the chaotic facies in Fig. 2b.

An area with a much lower FS value located at the continental slope east of Aceh Islands (marked as a white dashed line in
Fig. 6b) is also a reasonable landslide location for the tsunami simulation. However, this area is approximately 2.3 km east of the Aceh fault (the main SFZ segment), has a nearby large river system as a sediment deposit source, and no indication of chaotic facies on the seismic profile near to this location (Fig. 3); therefore, submarine landslides generating tsunamis are unlikely to occur here despite the low FS value at this location. However, this location was still used as a submarine landslide source for the tsunami simulation (location 8 in Fig. 6c). These evaluated marine sites (locations 1 to 8 in Fig. 6c)
are the possible locations of submarine landslides that are simulated in the tsunami scenarios described in the following section.

### 4.3 Tsunami model

To simulate a tsunami with an earthquake source, we must first distinguish the earthquake sources using a fault model. Genrich et al. (2000) suggested that the locking depth of the Aceh and Seulimeum faults does not exceed 15 km. The lengths
of the Aceh and Seulimeum faults on land are up to 200 and 120 km, respectively (Sieh and Natawidjaja, 2000); however, geophysical data are insufficient to reveal the details of their extensions to the northern offshore of Aceh. McCaffrey (1992) suggested that an earthquake of $M_w$ 7.5 or less is the largest possible event in the SFZ. The findings of Sieh and Natawidjaja (2000), who summarized available earthquake records, agree with this evaluation; accordingly, the maximum magnitude of an earthquake event was set as $M_w$ 7 in this study. This earthquake could occur at any location along the faults, but we chose
an epicenter location near the coast for our scenarios (star symbol in Fig. 6c). The magnitude, RA, RL, RW, $M_0$, $\overline{D}$, and focal mechanism of the proposed earthquake are summarized in Table 2. The earthquake focal mechanism in terms of the strike ($\theta$), dip ($\delta$), and slip ($\lambda$) were averaged from the Global CMT catalog (http://www.globalcmt.org/; Dziewonski et al., 1981; Ekström et al., 2012).

As presented in Fig. 6c and Table 2, two earthquakes were considered in the tsunami simulation. COMCOT was used to
compute the tsunami wavefields induced by both strike-slip earthquakes of $M_w$ 7 in the studied region, and tsunami waves with an amplitude of 0.3–0.5 m were generated in the entire coastal region. Thus, the tsunami hazard of a $M_w$ 7 strike-slip earthquake in this area (with epicentral distances less than 30 km) is negligible. Therefore, earthquakes generate strong ground motion, which triggers submarine landslides, but the tsunami hazard is due to the submarine landslide only. Typically, tsunamis triggered by submarine landslides have run-ups near the landslide location but have limited far-reaching effects.
The size of paleo-submarine landslides below the northern waters offshore of Sumatra in the tsunami simulation was estimated from seismic reflection data. The input parameters of COMCOT for submarine landslide sources were based on the collected information on regional active faults, the regional maximum earthquake magnitude, and the seafloor stability (Reid et al., 2015), and the parameters are listed in Table 2. A hypothetical submarine landslide with a length of 600 m and a width of 300 m (length–width ratio of 50%) was considered in this study. The COMCOT tsunami model indicated that this





submarine landslide with the aforementioned size and ratio could generate a tsunami with a wave height comparable to that in actual records, such as that of the recent Palu tsunami event studied by Gusman et al. (2019).

The computed spatial distribution of initial tsunami wave heights from eight submarine landslide sources (listed in Table 2) is presented in Fig. 7. All scenarios had depression waves toward shallow water, with subsidence ranging between 1.5 and 8.5 m; leading elevated waves propagated toward deep water, with rise ranging between 1.5 and 9 m. The largest initial wave
height was that in scenario 4, whereas the smallest was in scenario 1. However, scenario 3 had reversed polarity of the initial tsunami wave height compared with the other scenarios (Fig. 7). These deviations could be due to the landside location with respect to other morphological features, and the submarine landslide parameters were critical for generating the initial tsunami wavefields.

After evaluation of all the simulated tsunami scenarios, scenarios 4 and 7 are included in the discussion (other scenarios are
presented in the Supplementary Material). Figure 8 displays snapshots of simulated tsunami propagation for a submarine landslide at location 4 (Fig. 6c). The simulation reveals that a continental slope landslide at this location can generate a tsunami as high as 4 m; at several sites of the coast, the maximum amplitude of a tsunami wave can reach 8 m. These sites are located southwest of Weh Island (KW, Fig. 6c)—the closest coast to the landslide source. The propagation speed of the computed tsunami is affected by the water depth. The time evolution of a tsunami wave between 10 and 40 min after the
submarine landslide indicates higher speeds to the north of the source (north offshore of Krueng Raya); however, the amplitude is lower than that of waves with a lower speed propagating in the west–southwest direction and approaching the north shore of Banda Aceh. The snapshots in Fig. 8 also indicate that the frequency of tsunami waves changes when the wave front reaches the shallow water near the northern coast of Banda Aceh. The tsunami wave reaches the entire north coast of Aceh, west coast of Aceh Islands, and south coast of Weh Island in the first 10 min. The high-frequency wave is
distributed throughout the shallow coastal waters within 40 min.

Images of the simulated tsunami wave from a submarine landslide at location 7 (Fig. 6c) are presented in Fig. 9. Several islands surround the submarine landslide source. The tsunami wavefields reveal that the maximum tsunami wave high is approximately 2 m along the coast of Banda Aceh and approximately 3.5 m along the southwestern coast of Weh Island. This landslide-generated tsunami wave is blocked and propagates with a low speed in the northwestern direction. The
propagation wavefield is unblocked in the northeastern direction, and the tsunami wave has a higher amplitude of approximately 3.5 m in the southwestern coast of Weh Island (KW, Fig. 6c) and approximately 2.5 m along the eastern coast of Aceh Islands (LB and LN, Fig. 6c). Furthermore, the tsunami fully sweeps the northern coast of Banda Aceh and the surrounding islands within 10 min after the submarine landslide.

The computed initial tsunami wavefields of all scenarios (Fig. 7) reveal that the induced tsunami waves will extend and hit
the coasts of nearby islands. The computed maximum tsunami wave amplitudes at eight coastal sites were obtained for all submarine landslide scenarios (Fig. 6c). Statistical analysis of the distribution of the tsunami wave height at the eight selected sites is presented in Fig. 10 as a box-and-whisker plot (Massart et al., 2005). Landslide sources located at the northeastern coast of Banda Aceh (i.e., scenarios 1 to 5) generate higher tsunami waves that hit the entire northern coast of



Aceh compared with tsunamis with sources located at the northwestern coast of Banda Aceh (i.e., scenarios 6 to 8), which
only affect the northern coast of Banda Aceh, the eastern coast of Aceh Islands, and the southwestern coast of Weh Island.
However, the tsunami wave generated has nonsignificant effects on the eastern coast of Banda Aceh (Krueng Raya). The
tsunami in scenario 4 had the greatest tsunami hazard of all analyzed scenarios.

## 5 Discussion

Tsunamis induced by giant megathrust earthquakes, such as the 2004 Sumatra–Andaman earthquake or the 2011 Tohoku
earthquake in Japan, and their mechanisms have been investigated (Araki et al., 2006; Liu and Zhao, 2018; Romano et al.,
2014; Sibuet et al., 2007; Wang and Liu, 2006). These disastrous tsunamis were induced by a significant co-seismic
deformation due to sudden, vertical seafloor movement in the entire source area. Typically, earthquakes due to strike-slip
fault movement are not associated with significant uplift of the seafloor or with tsunami generation. In this study, the
tsunami wavefield from a strike-slip earthquake with source parameters listed in Table 2 was computed. Consistent with
previous reports, our results indicate that vertical seafloor movement is limited, and the induced tsunami wave is less than
0.5 m throughout the coastal regions in the study area, and the contribution of the earthquake to tsunami generation can be
neglected. However, the source rupture of a strike-slip fault (Table 2) dominates the generation of ground motion through its
horizonal components and its shaking of the seafloor sediment. To verify the calculated ground motion induced by the fault,
we followed the evaluation procedure of Phung et al. (2020) to predict ground motion by using four global ground motion
prediction equations (GMPEs), which were developed for the global application as a part of the NGA-West2 project:
[ASK14]; [BSSA14]; [CB14]; [CY14] (Abrahamson et al., 2014; Boore et al., 2014; Campbell and Bozorgnia, 2014; Chiou
and Youngs, 2014). The predicted ground motion is presented in Fig. 11. The predicted ground acceleration is greater than
0.4 g for epicentral distances less than 30 km. The eight landslide sites in Fig. 6c considered in this paper are all 30 km from
both simulated earthquakes, and the induced ground acceleration exceeds the pseudo-static acceleration threshold (0.14 g)
for triggering submarine landslides (Lee and Edwards, 1986).
According to computations conducted using the aforementioned proposed global GMPE models, the predicted ground
motion of a $M_w$ 7 strike-slip fault can exceed the pseudostatic acceleration threshold of 0.14 g for epicentral distances greater
than 70 km (Fig. 11). Thus, a $M_w$ 7 earthquake occurring on land may trigger a submarine landslide and may induce a large
tsunami. However, a submarine landslide–induced tsunami can be triggered by nearby small-magnitude offshore events.
Furthermore, multiple submarine landslides can be triggered by one event at failure sites on the continental slope, enhancing
tsunami hazards. The 2018 Palu earthquake is a real example of this phenomenon (Gusman et al., 2019).
Numerous examples have been presented indicating that submarine landslides can be triggered by earthquakes. To evaluate
the submarine landslide–induced tsunami hazard, detailed examination of the occurrence of large earthquakes is necessary.
Following the 2004 mega earthquake and tsunami, the seismic activity in northern Sumatra was low (Fig. 1b). The absence
of earthquakes in the northern SFZ (i.e., at the Aceh and Seulimeum faults) indicates that this region is vulnerable to future





earthquakes with a large magnitude (McCloskey et al., 2005; Nalbant et al., 2005). According to historical reports, a large event in 1936 ($M_w$ 7.1–7.3) (Newcomb and McCann, 1987; Sieh and Natawidjaja, 2000) seriously damaged the city of Banda Aceh. Harbitz et al. (2014) reported on historical tsunamis in Southeast Asia and described a much older event close to northern Sumatra. In 1837, this $M_w$ 7.3 event caused substantial damage at Banda Aceh and moderate damage at more

distant coastal areas, such as Penang Island (Malaysia) and Teluk Ayer (Singapore) (Harbitz et al., 2014; NGDC, 2019). However, source locations were not identified for either historical event. To capture this uncertainty in possible source locations, two strike-slip earthquakes with a magnitude as high as that of these historical events were simulated in offshore areas, and eight potential submarine landslide sites were considered in this study (Fig. 6c).

To mitigate tsunami threats from landslide sources, numerical modeling is a key method for both understanding the

landslides and predicting landslide-induced tsunamis (Harbitz et al., 2014; Masson et al., 2006). Modelling results indicated that response times in the northern tip of the Sumatra are less than 10 min for all evaluated scenarios. Due to the mechanism discrepancies of submarine landslides, the established Indonesian Tsunami Early Warning System (INATEWS), which was constructed to provide warnings for tsunamis induced by earthquakes further away, has limited capability to monitor submarine landslide–induced tsunamis. Therefore, a new tsunami hazard mitigation and early warning system for tsunamis

caused by landslides should be developed; this is crucial due to the evidence for a large MTD deposit with a clear sequence exposed in the seismic section. Although the escape buildings constructed during the rehabilitation and reconstruction following the 2004 disaster and some more of such building recently built by the government (personal communication), are ready for use, these buildings are still insufficient to accommodate the need for settlements from damaged coastal areas. This lack of refuge is another issue that must be overcome to successfully manage a submarine landslide tsunami event.

In this study, scenarios of submarine landslides triggered by a strike-slip fault earthquake, which would induce significant landslide tsunamis, have been demonstrated and quantitatively evaluated. Similar tectonic and environmental situations can be observed in other regions around the world, and our identified scenarios may also be relevant in these regions. However, the compounded tsunami and earthquake hazard in this submarine strike-slip fault system is still largely neglected in standard seismic hazard assessments.

**6 Conclusions**

A scenario of a submarine landslide in the northern waters offshore Sumatra triggered by a strike-slip fault system was proposed. The strike-slip fault movement in the marine environment was demonstrated to trigger a significant landslide on an unstable continental slope, inducing a tsunami; the effects were quantitatively evaluated. Evidence of a large MTD deposit was also observed in the northern offshore area. The northern tip of Sumatra has a high tsunami risk. This type of

tsunami can be triggered by a $M_w$ 7 earthquake occurring on land or by a nearby small-magnitude offshore event. Furthermore, multiple submarine landslides can be triggered by a single event, enhancing the tsunami hazard. Similar tectonic and environmental situations can be observed in other regions around the world, and our identified scenarios may

Natural Hazards
and Earth System
also be relevant in these regions. According to all scenarios evaluated in this study, near the coast, the warning time for the
landslide tsunami would be less than 10 min. The established Indonesian Tsunami Early Warning System constructed to
provide warnings for far earthquake-induced tsunamis has limited capability to monitor submarine landslide–induced
tsunamis. Further landslide tsunami hazard assessments and improvements in the early warning system in this area could be
achieved by using the proposed scenarios in this study.

**Data availability**

The data used in this study could be requested from the corresponding authors.

**Author contributions**

HAH and BSH initiated the original idea and conceptualized research. FK, CSLee, CSLiu, and AM performed the
processing and analysis on the single-channel seismic reflection data. HAH, CRW, SR, SP, IF, and IS performed the
processing and analysis on the Community-based bathymetric survey data. HAH and VBP performed the slope stability
analysis and ground motion prediction with input from BSH and WKL. HAH and IF performed the tsunami modelling with
input from TRW and BSH. All co-authors contributed to the interpretation of the results and to the manuscript writing led by
HAH and BSH.

**Competing interests**

The author declare that they have no conflict of interest.

**Acknowledgments**

The Body of Rehabilitation and Reconstruction Nanggroe Aceh Darussalam and Nias coordinated the CBBS project with
funding from the ADB (grant number 0002-INO Earthquake and Tsunami Emergency Support Project). This study was
supported by the Ministry of Education and Culture of the Republic of Indonesia under the Research Center of Universitas
Syiah Kuala within the H-Index Scheme (grant number 169/UN11/SPK/PNBP/2021). This study was also funded by the
Ministry of Science and Technology, Taiwan, grant MOST 108-2116-M-001-010-MY3 and grant MOST 109-2119-M-001-
011. Haekal A. Haridhi would like to thank the Taiwan International Graduate Program (TIGP), Academia Sinica, and
National Central University for sponsoring his Ph.D study. We thank Dr. Tedi Yudistira for his effort to digitize the SUMI
and SUMII single-channel seismic reflection data and Ifremer for access to this dataset, allowing us to fully use the results.
We thank Dr. Yu-Ting Kuo at the Institute of Earth Science, Academia Sinica, and Mr. Danny Chan at the Institute of
Geoscience, National Taiwan Ocean University, for their assistance in producing some of the figures. We thank Dr.



380 Syamsidik at the Tsunami and Disaster Mitigation Research Center, Syiah Kuala University for their support and discussion. We thank the Panglima Laot, fishermen of Lhok Krueng Aceh, the CBBS team (Syukri A., Bonar A., Dwi P., Arif F., Safrizal P., Nazaruddin B., Anjar S., Maria U., Siska M., Suraiya N., Putra D., Aris M., and Zariansyah) and the Network of Aquaculture Centers in Asia-Pacific for their cooperation, efforts, and support during the CBBS data collection. The Generic Mapping Tools software package was used to draw map figures (Wessel and Smith, 1991). Thanks for English edited by
385 Wallace Academic Editing.

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






**Table 1.** Slope stability analysis parameters

| Topography resolution (m) | | |
|---|---|---|
| Horizontal | Minimum | Maximum |
| 20 | -907 | 566 |

| Subsurface condition | | | | | |
|---|---|---|---|---|---|
| Material Properties | Groundwater Configuration | Earthquake Loading | Material Properties | | |
| | | | Cohesion | Angle of Internal Friction (°) | Weight |
| Homogeneous | None | 0.14 | 0 kPa | 26 | 17.5 kN/m3 |

| Stability Analysis | |
|---|---|
| Limit-Equilibrium Method | Search Method |
| Bishop's Simplified Limit Equilibrium | Box |



**Table 2.** COMCOT input parameter

| COMCOT Parameter | | | | | | | | |
|---|---|---|---|---|---|---|---|---|
| Fault Model (i.e. Strike-Slip Earthquake $M_w$ 7) | | | | | | | | |
| Depth (m) | RL (m) | RW (m) | $\overline{D}$ (m) | θ (°) | δ (°) | λ (°) | Epicenter at Aceh Fault (°) | Epicenter at Seulimeum Fault (°) |
| 10,000 | 58,884.4 | 12,882.5 | 1.50 | 330 | 89 | 130 | 95.227 E 5.612 N | 95.41 E 5.70 N |

| Submarine Landslide Scenario* | | | | | | | | |
|---|---|---|---|---|---|---|---|---|
| Parameter\case | 1 | 2 | 3 | 4 | 5 | 6 | 7 | 8 |
| c (°) | 95.49 E 5.65 N | 95.45 E 5.66 N | 95.40 E 5.70 N | 95.36 E 5.695 N | 95.33 E 5.66 N | 95.295 E 5.645 N | 95.27 E 5.6 N | 95.20 E 5.65 N |
| l (m) | 600 | 600 | 600 | 600 | 600 | 600 | 600 | 600 |
| w (m) | 300 | 300 | 300 | 300 | 300 | 300 | 300 | 300 |
| h (m) | 25 | 25 | 25 | 12.5 | 25 | 25 | 25 | 25 |
| φ (°) | 10 | 4 | 10 | 8 | 5 | 7.5 | 7.5 | 30 |

*The submarine landslide locations (scenario number 1-8) is shown in Fig. 6c.

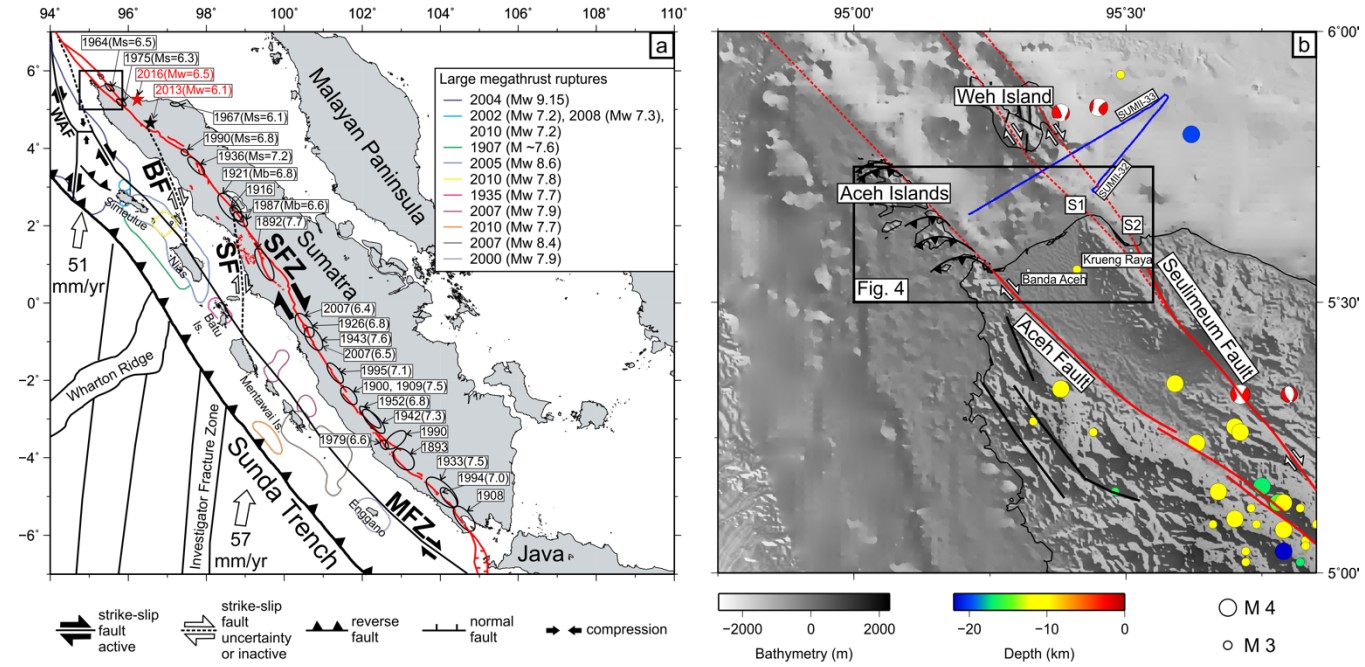





**Figure 1. (a)** Tectonics of the Sumatra subduction zone (Berglar et al., 2017). **(b)** Structure of the SFZ at northern Sumatra according to the interpretation of Fernández-Blanco et al. (2016). Recorded earthquake epicenters (color dots) and focal mechanisms were collected from the Badan Meteorologi, Klimatologi dan Geofisika (BMKG) of Indonesia and from the CMT global catalog (http://www.globalcmt.org/), respectively.

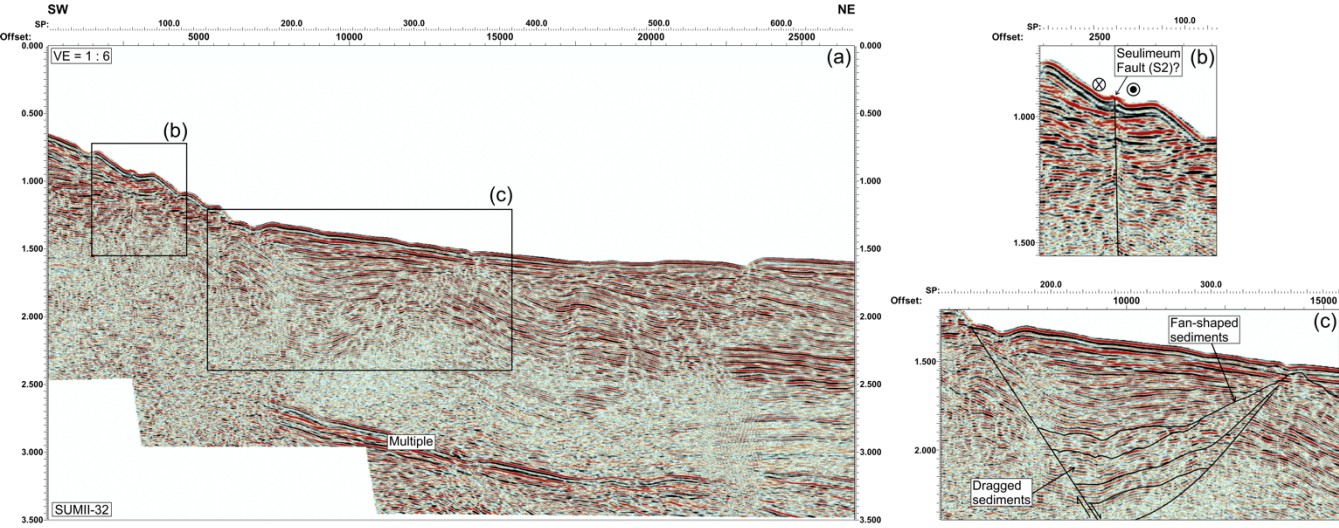


**Figure 2.** Seismic section of SUMII-32. (a) Uninterpreted seismic profile with direction, shot point (SP), and offset (in meters) presented at the top of the profile. (b) Fan-shaped sediments. (c) Possible location of the Seulimeum fault.



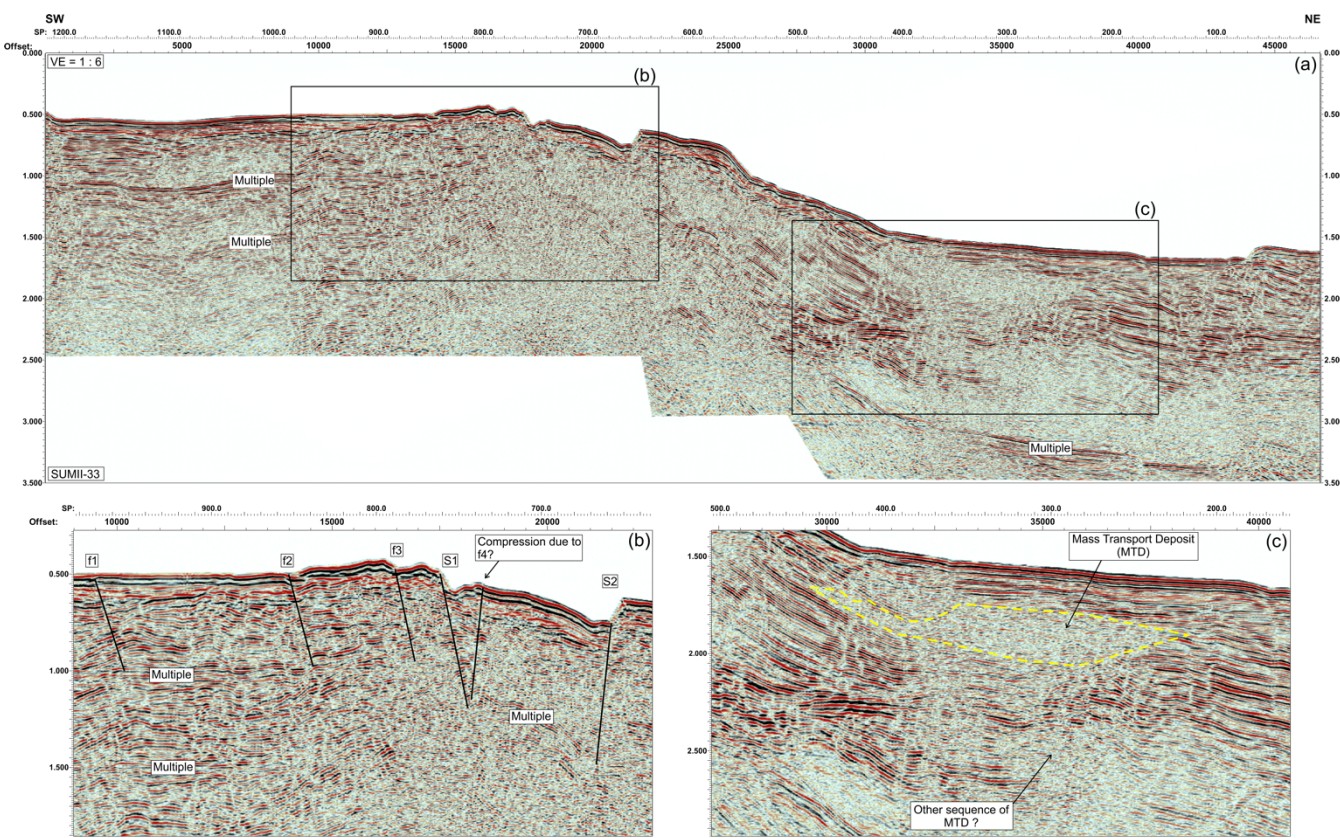

**Figure 3.** Seismic section of SUMII-33. **(a)** Uninterpreted seismic profile with direction, shot point (SP), and offset (in meters) presented at the top of the profile. **(b)** Possible compression. **(c)** Mass transport deposits.




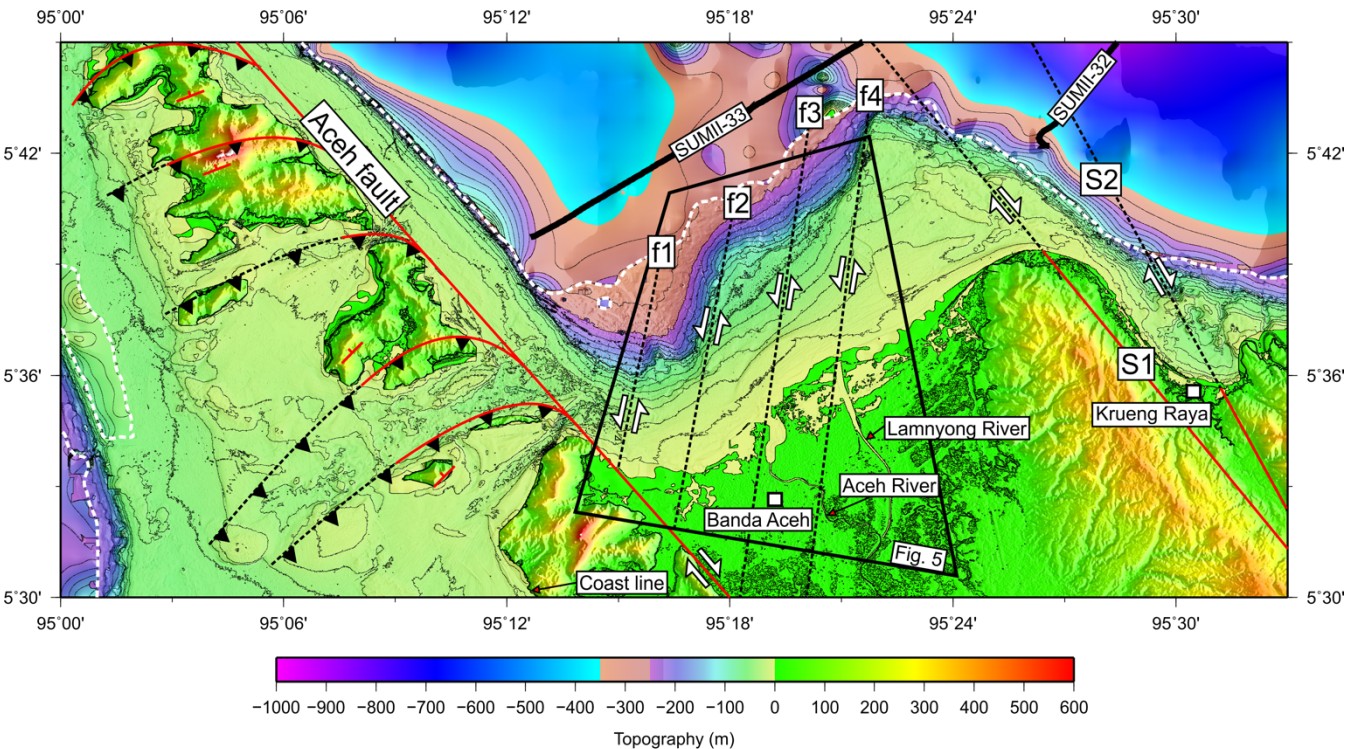

**Figure 4.** Topographic map of the northern tip of Aceh province. Figure 5 in a closeup of the trapezoidal box viewed from the northeast to southwest.





**Figure 5.** 3D topographic view of the northeastern shore and waters of Banda Aceh. The slope heights along f1 and f2 as well as the mound height and length are indicated. Colors enhance significant features along the continental slope.





**Figure 6.** Regional slope distribution, slope stability analysis, location of the submarine landslide sources for the tsunami simulation scenarios, and the location of near-coast cross sections. **(a)** Map of slope angles. **(b)** FS from the slope stability analysis; red indicates higher likelihood of failure. **(c)** Submarine landslide locations and their corresponding numbers; tsunami simulation scenarios are in Table 2.



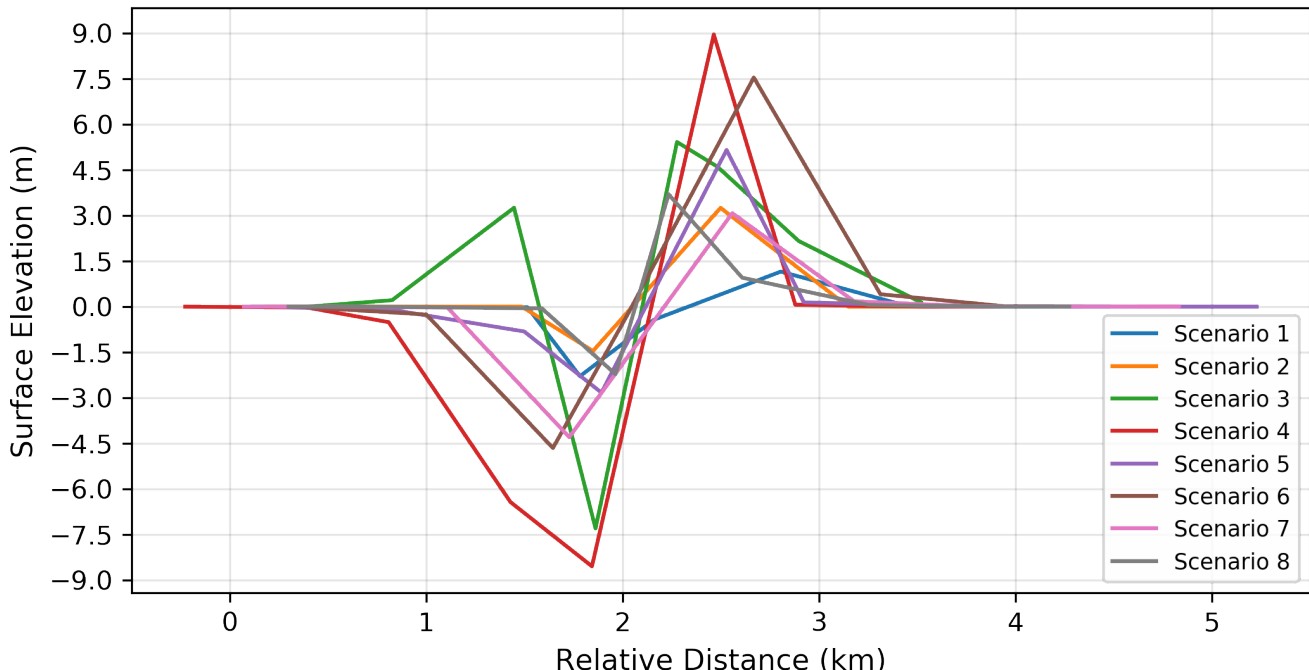

**Figure 7.** Initial tsunami wave heights for the eight submarine landslide scenarios.



**Figure 8.** Snapshots of a tsunami wave from a submarine landslide source at location 4 of Fig. 6c at propagation times of **(a)** 2 min, **(b)** 10 min, **(c)** 20 min, and **(d)** 40 min.





**Figure 9.** Snapshots of a tsunami wave from a submarine landslide source at location 7 of Fig. 6c at propagation times of **(a)** 2 min, **(b)** 10 min, **(c)** 20 min, and **(d)** 40 min.





**Figure 10.** Box and whisker plots (Massart et al., 2005) of the maximum tsunami wave amplitude along the selected coastlines of Fig. 6c. The "+" sign indicate the extreme values, while the solid red and dashed blue lines indicate the median and mean, respectively. The title of each plot indicates the cross-section location, the x-axis is shown for the landslide scenarios indicated in Table 2, while the y-axis indicates the tsunami wave height shown in meter.







**Figure 11.** Median response spectra predicted by the global candidate GMPEs for vertical strike-slip earthquakes with $V_{S30} = 310$ m s$^{-1}$ at selected epicentral distances ($R_X$). Sa names as abbreviation of spectral acceleration.