# Peer review of "Tsunami scenario triggered by submarine landslide offshore of northern Sumatra Island and its hazard assessment"

_Natural Hazards and Earth System Sciences, 2021_

## Author Comment (AC2)

**Supplementary material for**

**Tsunami scenario triggered by submarine landslide offshore of northern Sumatra Island and its hazard assessment**

Haekal A. Haridhi[1,9,10], Bor Shouh Huang[2, *], Kuo Liang Wen[3], Arif Mirza[4], Syamsul Rizal[1,9], Syahrul Purnawan[1,9], Ilham Fajri[12], Frauke Klingelhoefer[5], Char Shine Liu[4], Chao Shing Lee[6], Crispen R. Wilson[7], Tso-Ren Wu[8], Ichsan Setiawan[1,9,11] and Van Bang Phung[2]

[1]Department of Marine Sciences, Faculty of Marine and Fisheries, Universitas Syiah Kuala, Banda Aceh, Indonesia.

[2]Institute of Earth Science, Academia Sinica, Taipei, Taiwan.

[3]Department of Earth Sciences, National Central University, Taoyuan, Taiwan.

[4]Ocean Center, National Taiwan University, Taipei, Taiwan.

[5]Ifremer, Department of Marine Geosciences, Plouzané, France.

[6]Institute of Geosciences, National Taiwan Ocean University, Keelung, Taiwan.

[7]ENC, Washington, D.C., USA.

[8]Graduate Institute of Hydrological and Oceanic Sciences, National Central University, Taoyuan, Taiwan.

[9]Research Center for Marine Sciences and Fisheries, Universitas Syiah Kuala, Banda Aceh, Indonesia.

[10]Tsunami and Disaster Mitigation Research Center, Universitas Syiah Kuala, Banda Aceh, Indonesia.

[11]Graduate School of Mathematics and Applied Sciences, Universitas Syiah Kuala, Banda Aceh, Indonesia.

[12]Department of Capture Fisheries, Marine and Fisheries Polytechnic Aceh, Great Aceh, Indonesia.

*Correspondence to*: Bor Shouh Huang (hwbs@earth.sinica.edu.tw)

**Supplementary information:**

In line with our objectives to evaluate the possibility of the offshore extension of submarine environment of a strike-slip fault could produce a significant tsunami, the low Factor of Safety (FS) value at eastern offshore of Aceh island as it is shown in Fig. 6b, may also be seen as a candidate of a submarine landslide location and evaluate its capability to produce a tsunami. The location of the hypothetical submarine landslide as described in Table

2 and shown in Fig. 6c, are shown below (addition to the scenarios 4 and 7 that already shown in Figs. 8 and 9):

- Submarine Landslide Source Location 1 (Scenario 1)

[Figure]

Figure S.1 Snapshot of tsunami wave from a submarine landslide source at location 1 of Fig. 6c, at propagation times: (a). 2 minutes, (b). 10 minutes, (c). 20 minutes and (d). 40 minutes.

[Figure]

Figure S.2 Maximum tsunami wave amplitude from the corresponding source in Fig. S.1.

- Submarine Landslide Source Location 2 (Scenario 2)

[Figure]

Figure S.3 Snapshot of tsunami wave from a submarine landslide source at location 2 of Fig. 6c, at propagation times: (a). 2 minutes, (b). 10 minutes, (c). 20 minutes and (d). 40 minutes.

[Figure]

Figure S.4 Maximum tsunami wave amplitude from the corresponding source in Fig. S.3.

- Submarine Landslide Source Location 3 (Scenario 3)

[Figure]

Figure S.5 Snapshot of tsunami wave from a submarine landslide source at location 3 of Fig. 6c, at propagation times: (a). 2 minutes, (b). 10 minutes, (c). 20 minutes and (d). 40 minutes.

[Figure]

Figure S.6 Maximum tsunami wave amplitude from the corresponding source in Fig. S.5.

- Submarine Landslide Source Location 4 (Scenario 4)

[Figure]

Figure S.7 Snapshot of tsunami wave from a submarine landslide source at location 4 of Fig. 6c, at propagation times: (a). 2 minutes, (b). 10 minutes, (c). 20 minutes and (d). 40 minutes.

[Figure]

Figure S.8 Maximum tsunami wave amplitude from the corresponding source in Fig. S.7.

- Submarine Landslide Source Location 5 (Scenario 5)

[Figure]

Figure S.9 Snapshot of tsunami wave from a submarine landslide source at location 5 of Fig. 6c, at propagation times: (a). 2 minutes, (b). 10 minutes, (c). 20 minutes and (d). 40 minutes.

[Figure]

Figure S.10 Maximum tsunami wave amplitude from the corresponding source in Fig. S.9.

- Submarine Landslide Source Location 6 (Scenario 6)

[Figure]

Figure S.11 Snapshot of tsunami wave from a submarine landslide source at location 6 of Fig. 6c, at propagation times: (a). 2 minutes, (b). 10 minutes, (c). 20 minutes and (d). 40 minutes.

[Figure]

Figure S.12 Maximum tsunami wave amplitude from the corresponding source in Fig. S.11.

- Submarine Landslide Source Location 7 (Scenario 7)

[Figure]

Figure S.13 Snapshot of tsunami wave from a submarine landslide source at location 7 of Fig. 6c, at propagation times: (a). 2 minutes, (b). 10 minutes, (c). 20 minutes and (d). 40 minutes.

[Figure]

Figure S.14 Maximum tsunami wave amplitude from the corresponding source in Fig. S.13.

- Submarine Landslide Source Location 8 (Scenario 8)

[Figure]

Figure S.15 Snapshot of tsunami wave from a submarine landslide source at location 8 of Fig. 3.6c, at propagation times: (a). 2 minutes, (b). 10 minutes, (c). 20 minutes and (d). 40 minutes.

[Figure]

Figure S.16 Maximum tsunami wave amplitude from the corresponding source in Fig. S.15.

---

## Author Response (AR1)

**Answer to RC1**

We highly appreciate your time in reviewing the manuscript as well as your valuable comments. It is greatly improved the completness of our manuscript. Following please find the responses in detail:

Overall evaluation
Comment:

I think this is a good work and is helpful for tsunami hazards in Sumatra region. I recommend publication after the following revisions.

Answer:
We are glad that you are interested in our work as well as your positive feedback. Please find our corrections and responses to your comments and suggestions. The corrections are indicated in this response and shown in the marked-up manuscript version highlighting the changes (track changes in Word).

Comments:

1. Figure 1: Some text is Figure 1a cannot be read. Please increase fontsize.

Answer:
Thank you for your suggestion, we have revised by modifying Fig. 1a to increase the font size on some text. Well checked in electric version, it can be read without difficulty.

2. L25-28; what is meant by this: "In an extreme case, an earthquake of Mw 7 or more occurred, and the strong ground shaking triggered a submarine landslide off the northern shore of Sumatra."? Is this about a real event? Or hypothetical?

Answer:
Thank you for your question. We apologize that our explanation lacked clarity. It is a hypothetical event. We have provided clearer explanation. Following is the revised related text in the manuscript:

L25-28; "In an extreme hypothetical case, an earthquake of Mw 7 or more occurred, and the strong ground shaking triggered a submarine landslide off the northern shore of Sumatra." (see L26 of new revised MS)

3. Abstract: make your abstract more specific by adding some numbers and values from your modelling, like wave heights and etc.

Answer:
Thank you for your suggestion. We have revised the abstract and add the necessary information resulted from the modelling, such as wave heights. We revised the abstract as follow:

**Abstract.** Near the northern border of Sumatra, the right-lateral strike-slip Sumatran Fault Zone splits into two branches and extends into the offshore, as revealed by seismic sounding surveys. However, due to its strike-slip faulting characteristics, the Sumatran Fault Zone's activity is rarely believed to cause tsunami hazards in this region. According to two reprocessed reflection seismic profiles, the extended Sumatran Fault Zone is strongly associated with chaotic facies, indicating that large submarine landslides have been triggered. Coastal steep slopes and new subsurface characteristics of submarine

landslide deposits were mapped using recently acquired high-resolution shallow bathymetry data. Slope stability analysis revealed some targets with steep morphology to be close to failure. In an extreme hypothetical case, an earthquake of $M_w$ 7 or more occurred, and the strong ground shaking triggered a submarine landslide off the northern shore of Sumatra. Based on a simulation of tsunami wave propagation in shallow water, the results of this study indicate a potential tsunami hazard from several submarine landslide sources triggered by the strike-slip fault system can generate a tsunami as high as 4 - 8 m at several locations along the northern coast of Aceh. The landslide tsunami hazard assessment and early warning systems in this study area can be improved on the basis of this proposed scenario. (see L28-30 of new revised MS)

4. L43-55: Another good example of tsunami from strike-slip event from this region is the event of March 2016. See reference below. You could add something like this: "Heidarzadeh et al. (2017) showed potential tsunami hazards from strike-slip events by analysing the tsunami from the Mw 7.8 strike-slip earthquake in the Wharton Basin".

Answer:
Thank you for your suggestion. We have add on the manuscript as your suggestion. We revised the concerned paragraph as follow:

Analysis of the $M_w$ 7.0 Haiti earthquake on 12 January 2010 revealed that an earthquake with strike-slip faulting can produce a significant tsunami. Typically, a strike-slip fault movement is not associated with uplift of the sea floor or tsunami generation. However, a combination of other factors can trigger a tsunami. For the Haiti earthquake, the tsunami waves seem to have been caused by coastal failure landslides (Poupardin et al., 2020 and references therein). Satellite images and ground photos reveal changes in the coastline following the earthquake (Hornbach et al., 2010). The Haiti earthquake is not unique. On 28 September 2018, a large tsunami hit the city of Palu following the $M_w$ 7.5 Sulawesi earthquake in Indonesia. This event also occurred along a strike-slip fault. A tsunami of that size is unlikely to have been generated through earthquake rupturing alone. The tsunami is thought to have been caused by underwater and subaerial landslides triggered by the earthquake (Gusman et al., 2019). The complex bathymetry of the Palu Bay may have also contributed to the generation of the tsunami (Socquet et al., 2019). Another evaluation of strike-slip earthquakes that have caused tsunami is the Mw 7.6, 1999 Izmit earthquake, where slumping resulted from the gravitative instability of active gliding masses as the source of tsunami generation are observed as the chaotic deposit in the basin of the Sea of Marmara (Gasperini et al., 2022; Zitter et al., 2012). Heidarzadeh et al. (2017) showed potential tsunami hazards from strike-slip events by analyzing the tsunami from the Mw 7.8 strike-slip earthquake in the Wharton Basin. Other well-known tsunamis, such as the 1998 Papua New Guinea abnormal tsunami (Heinrich et al., 2001; Kawata et al., 1999; Tappin et al., 1999) and the 22 December 2018 tsunami at Sunda Strait caused by a flank collapse of the Anak Krakatau Volcano (Heidarzadeh et al., 2020; Muhari et al., 2019; Patton et al., 2018; Syamsidik et al., 2020), were also induced by earthquake-triggered submarine landslides (Ye et al., 2020). (see L54-57 of new revised MS)

5. Figures 2 & 3: here we have two issues: the fonts are small; and please write the owner of the data in the caption; is that from Malod and Kemal, 1996? Please also write that they are digitized from paper versions. These two figures are key figures of the paper and you need to be very clear about them.

Answer:
Thank you for your question and suggestion. We apologize that our explanation lacked clarity in the caption, although we have clearly indicate this data on L89-91. As it is in the

caption of Figs. 2 and 3, Yes, the seismic profiles are from Malod and Kemal, 1996. We revised the concerned caption as follow:

**Figure 2.** Seismic section of SUMII-32 that have been collected from 1991 to 1992 by Malod and Kemal (1996). This dataset are digitized from paper recording that were scanned and converted to digital images. All seismic traces were digitized and converted into the SEG-Y format for reprocessing. Please see section 3.1 for detailed processing of this dataset. (a) The reprocess uninterpreted seismic profile with direction, shot point (SP), and offset (in meters) presented at the top of the profile. (b) Possible location of the Seulimeum fault. (c) Fan-shaped sediments.

**Figure 3.** Same as Fig. 2 but for SUMII-33. (a) The reprocess uninterpreted seismic profile with direction, shot point (SP), and offset (in meters) presented at the top of the profile. (b) Possible compression. (c) Mass transport deposits.

6. L93: what type of reprocessing? Please clarify.

Answer:
Thank you for your question. We apologize that our explanation lacked clarity. The reprocessing of these seismic sections is explained as follows:

Due to digital conversion, the original seismic data has uneven trace amplitude with low-frequency noise artifacts clearly seen on some parts of the profile, so the main purpose of reprocessing is to attenuate those noises, while some post-stack image enhancement methods was also applied to further improve the seismic image. The processing detail is as follows: after SEG-Y input, a low-cut filter (4-8Hz) was applied to attenuate the low frequency artifact. To remove the noise outside the data range, seafloor mute and bottom trace mute were picked and applied, followed by amplitude balancing and signal enhancement in both frequency domain (FXDECON) and FK domain (FKPOWER). After that, post-stack predictive gap deconvolution was applied to remove the reverberation and compress the wavelet. Finally, seafloor mute and bottom trace mute were reapplied before SEG-Y output.

In the manuscript we add:

L92: Those paper recordings were scanned and converted to digital images. All seismic traces were digitized and converted into the SEG-Y format for reprocessing. In the absence of any velocity information, these data were migrated using a water velocity of 1500 m s−1 to remove the effects of seafloor scattering. Due to digital conversion, the original seismic data has uneven trace amplitude with low-frequency noise artifacts clearly seen on some parts of the profile, so the main purpose of reprocessing is to attenuate those noises, while some post-stack image enhancement methods was also applied to further improve the seismic image. The processing detail is as follows: after SEG-Y input, a low-cut filter (4-8Hz) was applied to attenuate the low frequency artifact. To remove the noise outside the data range, seafloor mute and bottom trace mute were picked and applied, followed by amplitude balancing and signal enhancement in both frequency domain and FK domain. After that, post-stack predictive gap deconvolution was applied to remove the reverberation and compress the wavelet. Finally, seafloor mute and bottom trace mute were reapplied before SEG-Y output. The reprocessed seismic profiles are presented in Figs. 2 and 3. (see L105-112 of new revised MS)

7. L295: Another good ref here: Tsuji et al. (2011):

Answer:

Thank you for your suggestion. We added the suggested reference and revised the paragraph as follows:

Tsunamis induced by giant megathrust earthquakes, such as the 2004 Sumatra–Andaman earthquake or the 2011 Tohoku earthquake in Japan, and their mechanisms have been investigated (Araki et al., 2006; Liu and Zhao, 2018; Romano et al., 2014; Sibuet et al., 2007; Tsuji et al., 2011; Wang and Liu, 2006). (see L316 of new revised MS and the same as in references)

8. L170: Yes, it is true that COMCOT can model landslide tsunamis as well. Would be useful to add another reference here of other people who used COMCOT for landslide tsunamis. I recommend Heidarzadeh and Satake (2015):

Answer:
Thank you for your suggestion. We added the suggested reference and revised the concerned line as follows:

COMCOT can also be used to simulate tsunamis caused by landslides (Heidarzadeh and Satake, 2015; Liu et al., 1995; Wang, 2009) (see L187 of new revised MS and the same as in references)

9. L316: I think here you could cite one more article; I suggest Heidarzadeh et al. (2019):

Answer:
Thank you for your suggestion. We added the suggested reference and revised the concerned line as follows:

Furthermore, multiple submarine landslides can be triggered by one event at failure sites on the continental slope, enhancing tsunami hazards. The 2018 Palu earthquake is a real example of this phenomenon (Gusman et al., 2019; Heidarzadeh et al., 2019) (see L337 of new revised MS and the same as in references)

**Answer to RC2**

We highly appreciate your time in reviewing the manuscript as well as your valuable comments. Following please find the responses in detail:

Overall evaluation

Comment:
This paper presents original seismic and bathymetric data off north Sumatra, showing active tectonic features and possible landslides scars. This dataset combined with numerical modeling allows to carry out tsunami simulations for selected scenarios defined on steep slopes where factor of safety is computed as low. This paper deserves a publication provided a minor review a made to address some of the questions raised below.

Answer:
We are glad that your positive feedback. Please find our corrections and responses to your comments and suggestions. The corrections are indicated in this response and shown in the marked-up manuscript version highlighting the changes (track changes in Word).

Comments:

1. The form of the manuscript consists of a first part basically on data and methods used, then a following part describes the main results, and the final discussion addresses again some results (for instance on the ground motion prediction models). Thus, the authors could state more clearly at the end of introduction, how it is organized, and a number of repetitions could be avoided, especially at the beginning of section 4. Another plan could have been to first address the whole section on stability (including analysis of acceleration threshold), and then a whole section on tsunami models which is rather independent. Finally, the highlight on the need of an early warning system is very necessary and this paper provides elements supporting possible related initiatives.

Answer:
Thank you for your suggestion. We apologize that our explanation lacked clarity and we totally understand the important reason advised by the reviewer. Actually, the predicted ground motion was used only as to validate the pseudo-static acceleration threshold for triggering submarine landslides, thus it is part of discussion, not dedicated as part of the result. For the clarity, we state the organization of the manuscript at the end of introduction section. We revised the L56-63:

An earthquake with a strike-slip fault rupture could also trigger a landslide and induce a tsunami offshore of northern Sumatra. In this study, to investigate the potential tsunami hazard at the northern tip of Sumatra, seismic reflection data were used to identify evidence of past submarine landslides. We collected detailed shallow bathymetric data of the area beyond the coast. This high-resolution bathymetric data was used to identify the fault traces and to evaluate the possibility of slope failure along the continental slope. The possibility of a submarine landslide triggered by earthquake shaking was examined through an analysis of the continental slope stability, and a tsunami caused by the combination of the earthquake and the resulting submarine landslide was simulated. The results indicated the characteristics of a potential landslide-induced tsunami and its potential damage. The predicted ground motion as a possible validation on strong ground shaking could induced the submarine landslide is discussed in discussion section. A possible tsunami early warning plan for hazard reduction is also discussed in this paper. (see L72-73 of new revised MS)

**Comments on section 1 introduction**

2. In addition to Haiti in 2010, Palu in 2018, the review of the tsunamis produced by strike slip earthquakes may also mention the case of the Izmit, 1999 (Turkey) quake that probably triggered submarine slides, and a tsunami observed at several coastal points.

Answer:
Thank you for your suggestion. We have added the necessary information as your suggestion. We revised the concerned paragraph as follow:

Analysis of the $M_w$ 7.0 Haiti earthquake on 12 January 2010 revealed that an earthquake with strike-slip faulting can produce a significant tsunami. Typically, a strike-slip fault movement is not associated with uplift of the sea floor or tsunami generation. However, a combination of other factors can trigger a tsunami. For the Haiti earthquake, the tsunami waves seem to have been caused by coastal failure landslides (Poupardin et al., 2020 and references therein). Satellite images and ground photos reveal changes in the coastline following the earthquake (Hornbach et al., 2010). The Haiti earthquake is not unique. On 28 September 2018, a large tsunami hit the city of Palu following the $M_w$ 7.5 Sulawesi earthquake in Indonesia. This event also occurred along a strike-slip fault. A tsunami of that size is unlikely to have been generated through earthquake rupturing alone. The tsunami is thought to have been caused by underwater and subaerial landslides triggered by the earthquake (Gusman et al., 2019). The complex bathymetry of the Palu Bay may have also contributed to the generation of the tsunami (Socquet et al., 2019). Another evaluation of strike-slip earthquakes that have caused tsunami is the Mw 7.6, 1999 Izmit earthquake, where slumping resulted from the gravitative instability of active gliding masses as the source of tsunami generation are observed as the chaotic deposit in the basin of the Sea of Marmara (Gasperini et al., 2022; Zitter et al., 2012). Heidarzadeh et al. (2017) showed potential tsunami hazards from strike-slip events by analyzing the tsunami from the Mw 7.8 strike-slip earthquake in the Wharton Basin. Other well-known tsunamis, such as the 1998 Papua New Guinea abnormal tsunami (Heinrich et al., 2001; Kawata et al., 1999; Tappin et al., 1999) and the 22 December 2018 tsunami at Sunda Strait caused by a flank collapse of the Anak Krakatau Volcano (Heidarzadeh et al., 2020; Muhari et al., 2019; Patton et al., 2018; Syamsidik et al., 2020), were also induced by earthquake-triggered submarine landslides (Ye et al., 2020). (see L54-58 of new revised MS)

3. Moreover, there is no mention of landslide triggering in the area, after the 2004 earthquake. The accelerations must have been very large enough to trigger some slides in the area, and it should be commented.

Answer:
Thank you for your suggestion. We have added the necessary information as your suggestion. We revised the concerned paragraph as follow:

Analysis of the $M_w$ 7.0 Haiti earthquake on 12 January 2010 revealed that an earthquake with strike-slip faulting can produce a significant tsunami. Typically, a strike-slip fault movement is not associated with uplift of the sea floor or tsunami generation. However, a combination of other factors can trigger a tsunami. For the Haiti earthquake, the tsunami waves seem to have been caused by coastal failure landslides (Poupardin et al., 2020 and references therein). Satellite images and ground photos reveal changes in the coastline following the earthquake (Hornbach et al., 2010). The Haiti earthquake is not unique. On 28 September 2018, a large tsunami hit the city of Palu following the $M_w$ 7.5 Sulawesi earthquake in Indonesia. This event also occurred along a strike-slip fault. A tsunami of that size is unlikely to have been generated through earthquake rupturing alone. The

tsunami is thought to have been caused by underwater and subaerial landslides triggered by the earthquake (Gusman et al., 2019). The complex bathymetry of the Palu Bay may have also contributed to the generation of the tsunami (Socquet et al., 2019). Another evaluation of strike-slip earthquakes that have caused tsunami is the Mw 7.6, 1999 Izmit earthquake, where slumping resulted from the gravitative instability of active gliding masses as the source of tsunami generation are observed as the chaotic deposit in the basin of the Sea of Marmara (Gasperini et al., 2022; Zitter et al., 2012). Heidarzadeh et al. (2017) showed potential tsunami hazards from strike-slip events by analyzing the tsunami from the Mw 7.8 strike-slip earthquake in the Wharton Basin. Other well-known tsunamis, such as the 1998 Papua New Guinea abnormal tsunami (Heinrich et al., 2001; Kawata et al., 1999; Tappin et al., 1999) and the 22 December 2018 tsunami at Sunda Strait caused by a flank collapse of the Anak Krakatau Volcano (Heidarzadeh et al., 2020; Muhari et al., 2019; Patton et al., 2018; Syamsidik et al., 2020), were also induced by earthquake-triggered submarine landslides (Ye et al., 2020). The recent mega earthquake such as the 2004 $M_w$ 9.2 Indian Ocean Tsunami and the 2011 $M_w$ 9.0 Tohoku earthquake may also include the submarine landslide as part of the tsunami source beside the major thrust fault movement, as the evidence of submarine landslide were observed for both earthquakes (Sibuet et al., 2007; Song et al., 2005; Tappin et al., 2014). (see L64-66 of new revised MS)

4. l.34: the details of the largest M 7.7 earthquake should be more explicit in the text. Was it the one that occurred in 1892? Another one in 1943 seems also to be of same importance.

Answer:
Thank you for your suggestion. We have added the necessary information as your suggestion. We revised the concerned paragraph as follow:

"large earthquakes have occurred along this long fault zone; the largest with a magnitude of 7.7 that occurred in 1892 at Angkola segment and a significant event with magnitude of 7.4 that occurred in 1943 at Sumani segment (Sieh and Natawidjaja, 2000) (see L35-36 of new revised MS)

5. l.52-55: the Dec 2018 tsunami in the Sunda Strait was not triggered by an earthquake-triggered event (l.55), but by the Krakatau volcano collapse. This Dec 2018 example is thus a good example of volcano-triggered tsunami, following a flank collapse. The end of the sentence l.55 should be modified accordingly, or the whole section.

Answer:
Thank you for your suggestion. We have modified the concerned line as your suggestion. We revised the concerned paragraph as follow:

Other well-known tsunami, such as the 1998 Papua New Guinea abnormal tsunami (Heinrich et al., 2001; Kawata et al., 1999; Tappin et al., 1999) was also induced by earthquake-triggered submarine landslides, while, the 22 December 2018 tsunami at Sunda Strait caused by a flank collapse of the Anak Krakatau Volcano (Heidarzadeh et al., 2020; Muhari et al., 2019; Patton et al., 2018; Syamsidik et al., 2020), which is a good example of volcano-triggered tsunami (Ye et al., 2020). (see L58-62 of new revised MS)

**Comments on section 2 Tectonic setting of the northern SFZ**

6. The setting is well explained and illustrated by Figure 1. It could be interesting to comment on the focal mechanisms plotted on the Figure 1, exhibiting diverse rupture modes for selected quakes (which magnitudes?), in the general complex tectonic frame of the area

Answer:
Thank you for your suggestion. To accommodate your suggestion, we revised the concerned paragraph as follow:

A recent detailed investigation of the Aceh and Seulimeum fault geometries revealed a complex fault system for both faults (Fernández-Blanco et al., 2016). The complex fault system at these fault segments are also reflected trough diverse rupture modes of the recorded earthquake focal mechanism (Fig.1b), which includes oblique right lateral strike-slip, with a complicated nodal plane. (see L87-89 of new revised MS)

**Comments on section 3 Collected data and analysis methods**

**3.1 Single-channel seismic reflection data**

7. Figure 2: captions (b) and (c) have to be switched

Answer:
Thank you for your correction and we apologize for the oversight on our part. We have revised the caption of Fig. 2b and c as follow:

**Figure 2.** Seismic section of SUMII-32 that have been collected from 1991 to 1992 by Malod and Kemal (1996). This dataset is digitized from paper recording that were scanned and converted to digital images. All seismic traces were digitized and converted into the SEG-Y format for reprocessing. Please see section 3.1 for detailed processing of this dataset. (a) The reprocess uninterpreted seismic profile with direction, shot point (SP), and offset (in meters) presented at the top of the profile. (b) Possible location of the Seulimeum fault. (c) Fan-shaped sediments. (see L595-599 of new revised MS)

**3.2 Community-based bathymetric survey data**

8. The Community-Based Bathymetric Survey was an interesting initiative following the 2004 tsunami to collect data from fishing boat tracks, that allowed to build a 20 m resolution bathymetric grid. This reveals four shear faults associated with significant scarps possibly associated with historical landslides. Is it this grid which is directly used later in Comcot models?

Answer:
Thank you for your question. It is ideal to use a high grid resolution for COMCOT models; however, it is very costly. The grid spacing that was used for the models on the first layer is 460 m or 0.25 arc-minutes and on the second layer a grid size ratio of 3 or about 154 m grid being applied.

**3.3 Slope stability analysis and input parameter assessment**

9. The stability analysis is modelled with the Scoops3D tool assuming uniform earthquake loading, and with parameters taken from other contexts (New Jersey, California). Maybe it could be commented how trustful these comparisons are, or not, and how they can be applied to north Sumatra. Are there any uncertainties that could be influent?

Answer: Thank you for your comment of the stability analysis is modelled with the Scoops3D. The computer program, Scoops3D, evaluates slope stability throughout a digital landscape represented by a digital elevation model (DEM). As shown in Table 1, limited reginal material properties within a user-defined size range have been applied in this computation. Without extra regional information, in this study, we employed previous

report material properties from some seismic active offshore margins (Lee and Edwards (1986) similar to the north Sumatra as inputs. It should be the best selection of this study. Of course, the uncertainties should be taken into account and reservation of our results.

**3.4 Simulation of tsunami wave propagation from earthquake and landslide sources**

10. The well-established COMCOT model is used for tsunami simulation. The authors should specify more clearly the bathymetric grid and the numerical parameters that are used, on top of the seismological parameters in Table 2. Has it been used with the 20-m bathymetric grid, or was it too costly to run it which such an accuracy?

Answer:
Thank you for your suggestion and question. We apologize that our explanation lacked clarity. As it is indicated in answer response no. 8, the 20 m grid was too costly to models. We have provided clearer explanation on the numerical parameters that are used as follow:

COMCOT can also be used to simulate tsunamis caused by landslides (Liu et al., 1995; Wang, 2009). In this study, we set a $1^{st}$ layer grid with 0.25 minutes resolution, and grid size ratio of 3 to the $1^{st}$ layer or about 154 m grid being applied to the $2^{nd}$ layer that both actives for the tsunami simulation. (see L188-189 of new revised MS)

11. Second, the landslide hypothesis implies the use of rigid block as the source. It should be stated that it is a very maximizing approach since a real landslide is more like a submarine deforming avalanche, much more complex to simulate than a simple rigid block. The Manning coefficient probably does not influence the results at the same order, or it should be more clearly explained and quantified.

Answer:
Thank you for your suggestion, we apologize that our explanation lacked clarity. We understand that the application of a rigid block as the source of landslide are far from reality, however, the limited resources to compute a complex deforming avalanche as well as it is a costly process made our option limited to use the rigid box scheme. We add the necessary information as suggested by reviewer such that there is a high chance that the result shown is overestimate the actual submarine landslide, we revised on the concerned text as follow:

L172:
Typically, modeling the time evolution of an actual landslide with seafloor changes requires substantial computations involving the detailed knowledge of local marine geological features and the landslide's triggering mechanism. Therefore, the model in this study used the rigid body movement as the source of submarine landslide are far from reality and could be overestimate the actual conditions. (see L192-194 of new revised MS)

12. Is the initiation computed in 2D horizontal coordinates or simply in 1D XZ section?

Answer:
Thank you for your question. The initiation was computed in 2D horizontal coordinates.

**Comments on section 4 Analysis and results**

**4.1    Evidence of paleo-landslides**

13. This section takes up some points presented in section 3 to recall the S2 fault activity, associated with chaotic facies possibly linked to landslide deposits. It is said that the bathymetric resolution is too low to localize the landslide site, however the 20 m x 20 m resolution mentioned earlier should theoretically help. Or is it insufficient because the original data are much sparser and interpolation makes them inadequate? In addition, there is at least one mound identified on Figure 5 while it is stated l.196 that any evidence of mound type structure is limited. It should be reformulated.

Answer:
Thank you for your question. We apologize that our explanation lacked clarity. We have provided clearer explanation on this subsection. The explanation is as follow: The 20 m x 20 m resolution of the CBBS data is located at continental shelf and slope, its furthest coverage is marked by the white dashed line on Fig. 4 and labeled on Fig. 5 "extend of CBBS dataset", thus, the bathymetric data at the seismic line (SUMII-32 and -33) where the chaotic facies is detected has a lower resolution and did not allowed to identify the landslide deposit. The statement on L196 "any evidence of mound type structure is limited" this means for the bathymetric features on the seismic line locations as stated on L195 "The low-resolution bathymetry data of the seismic survey". We revised the concerned line as follow:

L194-195:
However, the precise landslide site along the S1 and S2 faults are difficult to identify due to the low resolution of the obtained seafloor morphology data, as it is outside the height resolution CBBS bathymetric data coverage (Fig. 4). (see L215 of new revised MS)

**4.2 Stability evaluation of seafloor morphology**

14. The correlation between the slope stability and the seismic data is summarized on Figure 6, allowing to define possible landslide sources. The area east of the Aceh islands displays a very low FS and indeed it has to be considered. The fact that no landslide deposit has been identified there is not against the chance of having one triggering, following a large quake on the active branch nearby.

Answer:
Thank you for your suggestion. We totally agree that the chances of the landslide source with low FS indeed need to be considered. We modeled the tsunami at this location and the result is shown in Supplementary material submarine landslide source location 8 (scenario 8).

Below is the model result:

[Figure]

Figure S.15 Snapshot of tsunami wave from a submarine landslide source at location 8 of Fig. 3.6c, at propagation times: (a). 2 minutes, (b). 10 minutes, (c). 20 minutes and (d). 40 minutes.

[Figure]

Figure S.16 Maximum tsunami wave amplitude from the corresponding source in Fig. S.15.

15. By the way, it would be interested to define the equivalent earthquake magnitude needed to obtain the acceleration thresholds defined, depending on the distance to the rupture and ground motion prediction equation. A very large rupture on the Aceh fault is probably sufficient also to trigger distant destabilization.

Answer:
Thank you for your suggestion. This information could be seen on Fig. 11, where the acceleration threshold of 0.14 g for triggering submarine landslides could be achive at distance 70 km, and we also indicate this information in L312 – 317 "According to computations conducted using the aforementioned proposed global GMPE models, the predicted ground motion of a $M_w$ 7 strike-slip fault can exceed the pseudostatic acceleration threshold of 0.14 g for epicentral distances greater than 70 km (Fig. 11). Thus, a $M_w$ 7 earthquake occurring on land may trigger a submarine landslide and may induce a large tsunami. However, a submarine landslide–induced tsunami can be triggered

by nearby small-magnitude offshore events. Furthermore, multiple submarine landslides can be triggered by one event at failure sites on the continental slope, enhancing tsunami hazards. The 2018 Palu earthquake is a real example of this phenomenon (Gusman et al., 2019)."

**4.3    Tsunami model**

16. Two earthquakes with magnitude 7.0 are considered in the tsunami model, but the obtained amplitude of 0.3 to 0.5 m are considered as negligible (l. 246). The authors should specify if these heights are obtained at the coastal level or offshore. A coastal 0.3-0.5 m tsunami is not strictly negligible; it corresponds to the first degree of warning, allowing large debris and vulnerable people to be washed away. In addition, it could be also interesting to have a 7.5 scenario as a possible worst-case.

Answer:
Thank you very much for your suggestion. We understand that larger magnitude of earthquake could generate a larger ground shaking thus could have generated larger tsunami amplitude. However, the scenario presented in this study was set to mimic the possible condition of earthquake occurrence at the northern segment of SFZ as suggested by Nalbant et al. (2005), and the minimum magnitude indicated that likely to be occur in the future is $M_w7$. As a scenario, we use the $M_w$ 7 as a good threshold point to address the possible destruction that could be occurred, and indicate that the need of monitoring system as well as increase awareness to the society.
Thank you for your suggestion and we apologize for our lack of information we provided on the alert level of tsunami. We revised the text as follow:

L246:
Thus, the tsunami hazard of a $M_w$ 7 strike-slip earthquake in this area (with epicentral distances less than 30 km) is correspond to the first degree of warning or on the alert level (Badan Metereologi Klimatologi dan Geofisika, 2012). (see L266-267 of new revised MS)

Reference:
Nalbant, S. S., Steacy, S., Sieh, K., Natawidjaja, D. and McCloskey, J.: Earthquake risk on the Sunda trench, Nature, 435, 756–757, doi:10.1038/nature435755a, 2005.

17. Landslide scenarios using rigid block produce much higher amplitudes than for these earthquake scenarios, as displayed on Figure 7. The caption should mention that the distance of the X-axis is towards offshore to the right. The scenario 3 exhibits a quite different behavior with a relative peak towards the shore: this could be indeed explained by the localization of the slide, but it should be also explained, depending how the cross section is computed? Along a perpendicular to the coast? Or is the initiation purely in 1D?

Answer:
Thank you for your suggestion and question. We modify the caption on Fig. 7 to follow reviewer suggestion. The cross section is computed along a perpendicular to the coast, we add this information at L258.

Revision on the caption Fig. 7:
**Figure 7.** Initial tsunami wave heights for the eight submarine landslide scenarios with X-axis is towards offshore to the right. (see L614 of new revised MS)

Revision to the text at L258 as follow:
The computed spatial distribution of initial tsunami wave heights from eight submarine landslide sources (listed in Table 2) is presented in Fig. 7 computed along a perpendicular direction to the coast. (see L278 of new revised MS)

18. Are the following results computed along the shore with refined models? The Figure 7 seems to indicate a spatial sampling of about 500 m in the initiation model. Was it the same sampling throughout the modeling?

Answer:
Thank you for your question. The grid spacing for the model on the 1$^{st}$ layer is 0.25 minutes or about 460 m grid and grid size ratio of 3 to the 1$^{st}$ layer or about 154 m grid being applied to the 2$^{nd}$ layer that both actives for the tsunami simulation. It is the same sampling throughout the modeling.

19. l.261: typo: landslide rather than landside

Answer:
Thank you for your correction. We apologise for oversights on our part. We revised the concerned line as follow:

L262:
"These deviations could be due to the landslide location with" (see L282 of new revised MS)

**Comments on section 5 Discussion**

20. In the discussion again, the earthquake source is considered as negligible, but the authors should be more cautious. As stated previously in the paper, landslide tsunamis produce high local run-up. Earthquake tsunamis, even with 0.5 m amplitudes, can produce such heights at larger distances. As in Palu, the consequences are thus due to the combined effect of earthquake and several landslides.

Answer
Thank you for your suggestion. We apologise for oversights on our part. We revised the concerned statement as follow:

L300:
Consistent with previous reports, our results indicate that vertical seafloor movement is limited, and the induced tsunami wave is less than 0.5 m throughout the coastal regions in the study area, and the contribution of the earthquake to tsunami generation will resulted on first degree of warning or on the alert level (Badan Metereologi Klimatologi dan Geofisika, 2012). (see L321-322 of new revised MS)

21. In this discussion, it could be also interesting to have a comparison between the numerical models used for these scenarios in Sumatra, and for those used in Palu, since the situation could be very similar. Has the COMCOT tool been used in Palu too? Which hypotheses have been considered in Palu?

Answer:
Thank you for your suggestion. We totally understand the important reason advised by the reviewer. Therefore, we would like to confirm that a similar situation to the Palu earthquake resulting tsunami could also happen in the northern segment of SFZ. Thus, in the discussion section, we confirm this as a real example "The 2018 Palu earthquake is a

real example of this phenomenon (Gusman et al., 2019; Heidarzadeh et al., 2019)". To compare both (scenario in this study and the Palu tsunami) events similarity it seems difficult, due to the limitation on data and time that we have. (see L337 of new revised MS)

22. The final discussion on the risk posed by such landslide-triggered tsunamis is very necessary since the awareness is indeed very poor, and monitoring systems are not very efficient to provide warning at short distance. The authors could stress that the preparedness also relies on a better awareness among the populations, including the proper reaction after a strong shaking. This requires for instance to conduct frequent drills among the population to practice self-evacuation, has it been set up in Indonesia since 2004?

Answer:
Thank you for your suggestion. We totally agree with your suggestion. We have made additional important information regarding the preparedness and awareness among the population. Post the Indian Ocean Tsunami, there have been frequent tsunami drill activities as well as a significant increase in awareness of disaster along the northern coast of Aceh and Aceh province. We revised the concerned paragraph as follow:

L339:
This lack of refuge is another issue that must be overcome to successfully manage a submarine landslide tsunami event. One of the most crucial action to reduce the significant damage and victims are to enhance better preparedness and awareness of tsunami disaster. Although on the northern coast of Aceh the tsunami preparedness in Aceh is at a good level (Syamsidik et al., 2021), it is important to enhance the preparedness and awareness of a group community such as schools, disabilities, and others, while opportunities to enhance the involvement of local institutions could be increased in the Disaster Risk Reduction (DRR) related activities. (see L360-364 of new revised MS)